# Changes in functional outcome after a first-time stroke: Data from a longitudinal study

Nipaporn Butsing[1]*, Nalinrat Thongniran[2], Jesada Keandoungchun[3]

**1** Ramathibodi School of Nursing, Faculty of Medicine Ramathibodi Hospital, Mahidol University, Bangkok, Ratchathewi, Thailand, **2** Nursing Services Division, Department of Medicine, Faculty of Medicine Ramathibodi Hospital, Mahidol University, Bangkok, Ratchathewi, Thailand, **3** Division of Neurology, Department of Medicine, Faculty of Medicine Ramathibodi Hospital, Mahidol University, Bangkok, Ratchathewi, Thailand

* nipaporn.but@mahidol.edu

## Abstract

### Background

Strokes are the major cause of disability. Functional recovery following an acute stroke is time-dependent and varies depending on several factors. This study aimed to investigate changes in functional outcomes after stroke from discharge to six months post-discharge and to identify factors associated with changes in functional outcomes during this six-month period.

### Methods

The study included 155 consecutive patients with a first stroke and were hospitalized in an acute stroke unit at an advanced tertiary hospital in Bangkok between April 2020 and August 2021. The patients' functional outcomes were evaluated using the modified Rankin Scale (mRS) and the Barthel Index (BI) before hospital discharge and at one-, three-, and six-month post-discharge. Friedman tests were used to assess changes in mRS scores over six months. Linear mixed effect regression was applied to identify the change in BI scores during the six months post-discharge.

### Results

Of the participants, 62.6% were men, and the mean age was 64.0 (SD = 12.5). The median mRS scores ranged from 3.0 at discharge to 0.0 at six months post-discharge. Significant changes in mRS scores were identified within three months post-discharge, and differences by stroke subtype and severity (P < 0.05). The crude BI means ranged from 70.6 (SD = 28.5) at discharge to 93.1 (SD = 20.4) at six months post-discharge. Female participants and those with hemorrhagic strokes had lower adjusted BI scores compared to male participants and those with ischemic strokes, with differences of 4.1 and 4.5 points, respectively. Additionally, stroke severity was

**Data availability statement:** All relevant data is included in the manuscript and the Supporting information file. Unfortunately, raw data cannot be shared publicly due to the institution's data-sharing policy, as it contains patients' clinical information. However, data is available upon request and subject to ethics committee approval. Long-term data storage and availability are ensured under the project approved by the institution's ethics committee (COA No. MURA2024/597). Researchers interested in accessing the data may submit their research proposals to the Human Research Ethics Committee at the Faculty of Medicine, Ramathibodi Hospital, by visiting https://www.rama.mahidol.ac.th or by contacting them via email at raec.mahidol@gmail.com or by telephone at +66 22012772. Researchers can also reach out to the corresponding author for assistance.

**Funding:** The author(s) received no specific funding for this work.

**Competing interests:** The authors have declared that no competing interests exist.

inversely related to adjusted BI scores. One increased National Institute of Health Stroke Scale (NIHSS) score decreased adjusted BI scores by 3.6.

## Conclusion

The time after discharge, gender, stroke subtype, and stroke severity are significant factors affecting functional outcomes after a stroke. The most significant improvement in functional outcomes occurred within one month post-discharge.

---

## Introduction

Stroke is one of the leading causes of death and disability worldwide [1]. In 2019, stroke caused 6.6 million deaths, and 143 million disability-adjusted life years (DALYs) lost, making it the world's second-leading cause of death and third-leading cause of disability [1]. In 2019, the stroke incidence was approximately 12.2 million [1]. The burden of stroke is expected to increase as populations age, including in Thailand [2]. Thailand has turned into an aging society [2] and stroke is a significant cause of mortality and disability [3]. In 2004, stroke caused the highest DALYs in Thai women (315,500 DALYs lost) and the third leading cause of DALY in Thai men (366,600 DALYs lost) [3]. A patient with ischemic stroke or hemorrhagic stroke was estimated to lose 5.5 or 10.7 years of life, respectively [4].

Medical technology advancements can reduce mortality and prolong patients' lives after stroke [5]. It was estimated that stroke survivors had an average life expectancy of 13.6 years [4]. Despite advancements in medical technology, more than half of stroke survivors suffer from neurological deficits [5–10]. Stroke consequences include neurological, cognitive, and functional impairment. The levels of impairment vary with location and severity of stroke [11,12]. Such consequences can lead to an economic burden, such as increased health expenditure and decreased productivity [13,14].

Stroke survivors frequently suffer functional disabilities that impede their everyday life and make them dependent on others [6–10]. Functional outcomes in stroke refer to the degree of disability of patients' daily activities or disability in performing activities of daily living [5–7,9,10,15]. The functional outcomes in stroke patients have been measured by the modified Rankin scale (mRS) [5,16,17] and the Barthel index (BI) [17–19]. These tools are simple measures of scoring functional recovery after a stroke [5,16,17,19]. The optimal recovery of stroke patients varies over the first few months and may continue up to six months [20]. Previous studies measured functional outcomes of stroke within three months and found significant changes at 1 month and 3 months after stroke [9,15]. The Australian New Zealand Clinical Trials study found some changes in mRS at six months, but the distribution of mRS scores seemed steady from six to twelve months after stroke [10].

According to studies on stroke recovery, the most prognostic factors for functional recovery are gender [6,15], age [11,21,22], stroke subtype [6,15,18], and stroke severity [6,9]. Some longitudinal studies limit functional outcome measurements to three months [9,15], which may not cover possible improvements in functional

outcomes in stroke survivors [10,20]. A previous study included only mild stroke patients [9]. This study attempted to explain how functional outcomes change within six months following a stroke and the impact of such prognostic factors on functional outcomes. Identifying patterns of functional improvement in stroke patients over time may help healthcare providers tailor appropriate care for patients based on their related factors.

The objectives of this study are 1) to describe modified Rankin scale and Barthel index scores at the time of discharge, at 1-month, 3-month, and 6-month post-discharge; 2) to identify changes in mRS score from the time of discharge to 1-month, 3-month and 6-month post-discharge by gender, stroke subtype and stroke severity; and 3) to identify factors (time, gender, age, stroke subtype and stroke severity) for changes in Barthel index score over the period of 6 months in patients with first stroke.

## Materials and methods

### Study sample

Data was accessed for this study on 1 September 2024. This study used the longitudinal data from a previous cohort study conducted between 1 April 2020 and 10 October 2021. The initial study screened 186 consecutive acute first-stroke patients hospitalized at an advanced tertiary hospital in Bangkok, Thailand. Out of 186 patients, 155 were eligible for the study based on the following inclusion criteria: 1) aged ≥ 20 years, 2) had a first ischemic or hemorrhagic stroke confirmed by brain imaging, 3) were capable of cognitive-communication, and 4) willingly agreed to participate in the study. The exclusion criteria included diagnosis of recurrent stroke or other chronic diseases (e.g., cancers, heart failure, end-stage renal diseases), language-related limitations (e.g., aphasia), and failure to give consent for the study.

The initial study protocol was approved by the Human Research Ethics Committee, Faculty of Medicine Ramathibodi Hospital (COA. MURA2020/193). All methods were performed in accordance with the principles outlined in the Declaration of Helsinki. Full disclosure was provided to eligible participants, and informed consent was obtained before data collection. The Human Research Ethics Committee, Faculty of Medicine Ramathibodi Hospital reviewed and approved this recent study to use anonymized longitudinal data from the initial project (COA. MURA2024/597, dated 23 August 2024).

In the main project, all recruited participants were interviewed, and their general characteristics, clinical characteristics, and functional outcomes were recorded the day before their hospital discharge. Then, telephone interviews for functional outcome assessments were conducted one month, three months, and six months after discharge. Some characteristics of the study samples were previously reported in an earlier study [23].

### Measures

**General and clinical characteristics.** The general characteristics of the study participants included age, gender, marital status, highest education level, body mass index (Kg/m$^2$), and monthly income sufficiency. Clinical characteristics included comorbidities, stroke subtype (ischemic stroke or hemorrhagic stroke), location of stroke, specific treatment, and severity of stroke. The National Institutes of Health Stroke Scale (NIHSS) scores on participants' discharge were used to iden- tify the remaining stroke severity before discharge. An NIHSS score of 0 indicates no stroke symptoms, a score of 1–4 indicates a minor stroke, a score of 5–15 indicates a moderate stroke, a score of 16–20 indicates a severe stroke, and a score of 21–42 indicates a very severe stroke [24].

**Functional outcomes.** The modified Rankin scale (mRS) and the Barthel index (BI) were used to evaluate the functional outcomes of the study participants. Scores on the mRS and BI were assessed at discharge, and at 1 month, 3 months, and 6 months post-discharge, to identify changes in the functional status of post-stroke patients.

The modified Rankin scale (mRS) is a commonly used tool to quantify the degree of disability in stroke patients [5,16,17]. This ordinal scale rates patients' disability in regular daily tasks from 0 (no stroke symptoms) to 6 (death). The mRS can be used to measure the functional outcomes of stroke patients in both the acute and rehabilitation periods

[5,16,18,22]. A mRS score of 1 indicated no significant disability, implying that patients can perform all routine tasks despite symptoms, while a score of 5 indicated the most severe level of disability, implying that patients require continual nursing care and attention [16,18].

The Barthel index (BI) is a scale that quantifies one's ability to carry out daily activities. It is widely used to assess functional disability and is the most commonly used functional measure in stroke rehabilitation settings [19]. BI is a 10-item scale that ranges from 0 to 100 in 5-point increments, with higher scores indicating greater independence in ADL [25]. The items in BI assess a patient's capacity in feeding, bathing, grooming, dressing, bowel control, bladder control, toilet use, chair transfer, mobility, and stair climbing [25].

This study followed the Strengthening the Reporting of Observational Studies in Epidemiology (STROBE) checklist for cohort study [26].

## Statistical analysis

This study included 155 first-stroke participants. All 155 participants had modified Rankin scores (mRS) and Barthel index (BI) at discharge and one-month post-discharge. One patient died from sepsis between one and three months after discharge, while another was lost to follow-up. Another patient was admitted to the ICU at six months post-stroke discharge due to heart failure (see Fig 1). As a result, mRS and BI scores were obtained from 155, 155, 153, and 152 stroke participants at discharge, one month, three months, and six months after discharge, respectively (Fig 1).

The IBM SPSS Statistics version 29.0 (IBM Corp, Armonk, NY) was used to analyze data. Categorical data were reported as frequencies and percentages. Continuous variables were presented as mean, standard deviation (SD), median, interquartile range (IQR), minimum, and maximum values. The distribution of modified Rankin scale scores

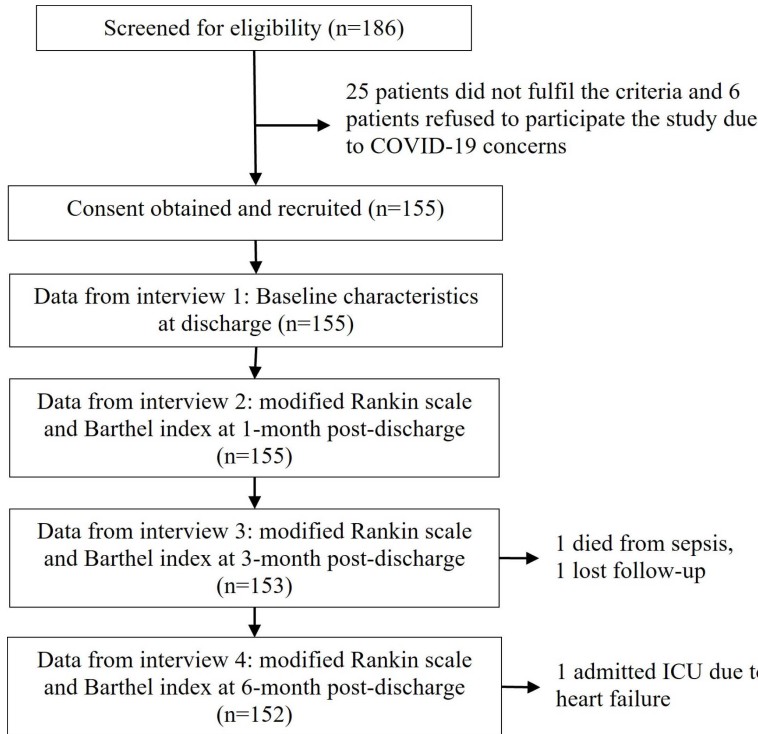

**Fig 1. Study sample.**

(by category) was provided together with figures that illustrate the patterns of functional states of stroke survivors over six months. The mRS ordinal scores were compared from the hospital discharge to 1 month, 3 months, and 6 months post-discharge using the Friedman variance test by ranks. The Friedman analyses of mRS rank scores were stratified by gender, stroke subtype, and stroke severity. Bonferroni correction was applied for post hoc comparisons to assess differences in mRS between two time points.

The mean (SD) of Barthel index scores at discharge, 1-month, 3-month, and 6-month post-discharge were analyzed by gender, stroke subtype, and stroke severity. Linear mixed effect models were used to assess changes in Barthel index scores at discharge, one month, three months, and six months after discharge. The impacts of gender, age, stroke subtype, and stroke severity on BI improvement were also investigated with linear mixed effect models [27]. The likelihood ratio (LR) tests were used to compare the full and reduced models for the mean BI scores, which included main effect variables. Statistical significance was determined by a p-value <0.05.

## Results

### Baseline characteristics of stroke participants

The study initially enrolled 155 stroke participants. The baseline characteristics are provided in Table 1. The average age was 64.0 years (SD = 12.5). Most participants (62.6%) were male, with 67.1% married. Approximately 52.9% completed primary or secondary school. Most of the participants were obese (38.3%), followed by overweight (16.9%) and very obese (14.9%). The most prevalent comorbidities among stroke participants were hypertension (64.5%), dyslipidemia (54.2%), and diabetes mellitus (38.7%). The majority of study participants (89.7%) suffered from ischemic stroke. The study participants had a variety of stroke lesion locations, including the left cerebral hemisphere (29.0%), the right cerebral hemisphere (12.9%), both cerebral hemispheres (0.6%), cerebellum (11.0%), thalamus (2.6%), basal ganglia (2.6%), brainstem (11.6%), and unspecified location (29.7%). The median NIHSS score of the study participants after treatment was 3.0 (IQR = 4.0). About half of the participants were discharged with minor strokes (51.0%), 23.9% with moderate strokes, 18.1% with no stroke symptoms, and 7.1% with severe strokes. More than half of the participants (53.6%) were discharged with moderate to severe disability (mRS ≥ 3). The mean Barthel index score at discharge was 70.6 (SD = 28.5).

### Patterns of changes in modified Rankin scores

Fig 2 presents the distribution of mRS scores during six months. There was a reduction in the severity of mRS scores over time. The highest proportions of mRS reduction were identified from discharge to 1-month, followed by 1-month to 3-month post-discharge, and small changes from 3-month to 6-month post-discharge. Over six months after discharge, ischemic stroke participants had a lower proportion of functional disability than hemorrhagic stroke participants. Considering patterns of mRS change by stroke severity, all stroke participants with no stroke symptoms reported no symptoms (mRS = 0) and no significant disability (mRS = 1) from 3 to 6 months after discharge. Patients who experienced minor strokes showed a reduction in disability over time; approximately 3.9% had a mRS score of 2 or higher at six months. For patients with a moderate stroke, the degree of disability decreased over time. However, the proportion of those with moderate to severe disability (mRS ≥ 4) had not changed from 3 months to 6 months post-discharge. Patients with a severe stroke had moderately severe disability (mRS = 4, 9.1%) and severe disability (mRS = 5, 90.9%) in their functional abilities at discharge. There was a very small reduction in the degree of disability over time; not all patients were free of disability six months after discharge (mRS ≥ 2). Patients with severe stroke remained in severe disability status in routine tasks (mRS ≥ 4) at six months after discharge (72.8%). Details shown in S1 Table.

Table 2 presents the results of the Friedman test and post hoc comparisons of median scores on the modified Rankin Scale (mRS) from discharge to one, three, and six months later. The mRS scores significantly decreased from hospital discharge to 6 months post-discharge (p < 0.05). The median mRS scores significantly decreased from discharge to 1 month (p < 0.05) and from 1 month to 3 months (p < 0.05), but no significant improvement was observed from 3 months

**Table 1. General characteristics and clinical characteristics of patients with stroke (n = 155).**

| Characteristics | Number (%) |
| --- | --- |
| **Age (years)**, mean (SD); min-max; median (IQR) | 64.0 (12.5); 29-91; 64.0 (17.0) |
| **Gender** | |
| Male | 97 (62.6) |
| Female | 58 (37.4) |
| **Marital status** | |
| Married | 104 (67.1) |
| Not married | 51 (32.9) |
| **Level of education** | |
| Primary school | 47 (30.3) |
| Secondary school | 35 (22.6) |
| Vocational certificate/Diploma | 15 (9.7) |
| Bachelor's degree | 40 (25.8) |
| Higher than bachelor's degree | 17 (11.0) |
| **BMI (Kg/m$^2$)**, mean (SD); min-max; median (IQR) | 25.7 (4.4); 15.8-39.8; 25.5 (6.5) |
| Underweight (<18.5) | 3 (1.9) |
| Normal weight (18.5–22.9) | 43 (27.9) |
| Overweight (23–24.9) | 26 (16.9) |
| Obese (25–29.9) | 59 (38.3) |
| Very obese (≥30) | 23 (14.9) |
| **Monthly income** | |
| Sufficiency income | 130 (83.9) |
| Insufficiency income | 25 (16.1) |
| **Comorbidities\*** | |
| Diabetes mellitus | 60 (38.7) |
| Hypertension | 100 (64.5) |
| Dyslipidemia | 84 (54.2) |
| Ischemic heart disease | 27 (17.4) |
| Atrial fibrillation | 25 (16.1) |
| Chronic kidney disease | 10 (6.5) |
| Chronic lung disease | 4 (2.6) |
| **Subtype of stroke** | |
| Ischemic | 139 (89.7) |
| Hemorrhagic | 16 (10.3) |
| **Location of stroke** | |
| Left cerebral hemisphere | 45 (29.0) |
| Right cerebral hemisphere | 20 (12.9) |
| Both cerebral hemispheres | 1 (0.6) |
| Cerebellum | 17 (11.0) |
| Thalamus | 4 (2.6) |
| Basal ganglia | 4 (2.6) |
| Brain stem | 18 (11.6) |
| Unspecified | 46 (29.7) |
| **Specific treatment** | |
| rTPA | 19 (12.3) |
| Mechanical thrombectomy | 11 (7.1) |
| Brain surgery | 1 (0.6) |

*(Continued)*

**Table 1.** (Continued)

| Characteristics | Number (%) |
|---|---|
| **NIHSS at discharge**, mean (SD); min-max; median (IQR) | 4.2 (4.7); 0-23; 3.0 (4.0) |
| No stroke (NIHSS 0) | 28 (18.1) |
| Minor stroke (NIHSS 1–4) | 79 (51.0) |
| Moderate stroke (NIHSS 5–15) | 37 (23.9) |
| Severe stroke (NIHSS 16–42) | 11 (7.1) |
| **mRS at discharge**, mean (SD); min-max; median (IQR) | 2.7 (1.7); 0-5; 3.0 (3.0) |
| No symptoms (0) | 16 (10.3) |
| No significant disability (1) | 38 (24.5) |
| Slight disability (2) | 18 (11.6) |
| Moderate disability (3) | 20 (12.9) |
| Moderately severe disability (4) | 33 (21.3) |
| Severe disability (5) | 30 (19.4) |
| **BI at discharge**, mean (SD); min-max; median (IQR) | 70.6 (28.5); 10-100; 75.0 (46.0) |

*Multiple responses.

BMI = Body mass index; NIHSS = National Institute Health Stroke Scale; mRS = modified Rankin Scale; BI = Barthel Index; SD = Standard deviation; min = minimum; max = maximum; IQR = Interquartile range.

to 6 months after discharge. Male stroke participants had lower degrees of disability than females. Participants who experienced hemorrhagic strokes had worse functional outcomes compared to those with ischemic strokes. There were significant changes in the median scores of the mRS from discharge to 1 month, 3 months, and 6 months after discharge among those with hemorrhagic stroke ($p < 0.05$); however, no significant changes were observed in scores from 1 month to 3 months after discharge. Participants with a more severe stroke had a lower functional recovery. Participants with minor strokes had significantly different mRS median scores in almost all pairs of time points ($p < 0.05$), except from 3 to 6 months following discharge. Participants with moderate strokes had significant improvement in mRS from discharge to 3 months and from 1 month to 3 months post-discharge ($p < 0.05$), but no significant change after 3 months. The median mRS scores for patients with severe stroke decreased from discharge to six months post-discharge. However, Bonferroni adjustment results showed no significant differences between any time points.

### Changes in Barthel index scores

Table 3 shows that the crude mean Barthel index (BI) scores increased from 70.6 (SD = 28.5) to 89.6 (SD = 23.8) in one month, 92.9 (SD = 19.8) in three months, and 93.1 (SD = 20.4) in six months after discharge. The mean BI score at discharge was lower in female stroke participants (mean = 63.0, SD = 28.9) than in male participants (mean = 75.1, SD = 27.3). The BI scores in males were also higher than in females at one, three, and six months after discharge; however, the mean differences were smaller (6.9, 3.3, and 4.4 scores, respectively) than at discharge (12.1 scores). The mean BI score at discharge among hemorrhagic stroke patients was lower (mean = 45.9, SD = 31.1) than those with ischemic stroke (mean = 73.4, SD = 26.8). The BI scores increased between discharge and 1 month, 3 months, and 6 months post-discharge for both stroke subtypes. The severe stroke participants had the lowest mean BI score (mean = 20.9) at the time of discharge, followed by moderate stroke participants (mean = 50.3), minor stroke participants (mean = 78.9), and those with no stroke symptoms at discharge (mean = 93.7). The mean BI scores showed considerable improvement for stroke participants with minor stroke and moderate stroke at 1 month, 3 months, and 6 months post-discharge. However, the mean BI score among severe stroke participants seemed to improve over time, but the scores were still low at six months post-discharge (mean = 48.5).

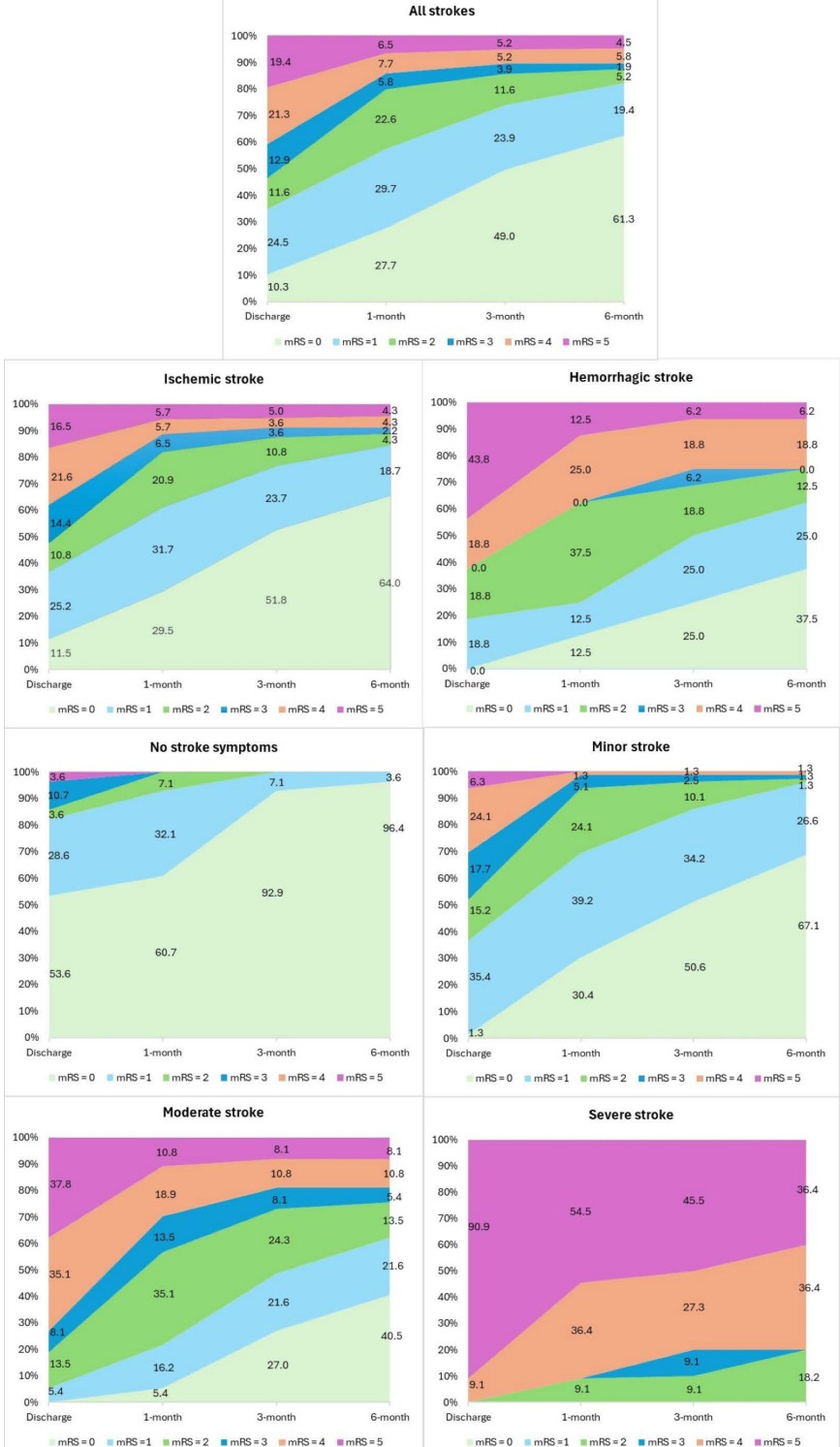

**Fig 2. Distribution of mRS scores at discharge (n = 155), 1-month (n = 155), 3-month (n = 153), and 6-month (n = 152) post-discharge.**

**Table 2. Changes of mRS scores at discharge, 1-month, 3-month, and 6-month post-stroke.**

| Variables | n | Discharge Median (IQR) | 1-month Median (IQR) | 3-month Median (IQR) | 6-month Median (IQR) | P-value | Post hoc-test* 0 1 | 0 3 | 0 6 | 1 3 | 1 6 | 3 6 |
|---|---|---|---|---|---|---|---|---|---|---|---|---|
| **All** | 152 | 3.0 (3.0) | 1.0 (2.0) | 0.5 (2.0) | 0.0 (1.0) | <0.001 | * | * | * | * | * | |
| **Gender** | | | | | | | | | | | | |
| Male | 95 | 2.0 (3.0) | 1.0 (2.0) | 0.0 (1.0) | 0.0 (1.0) | <0.001 | * | * | * | * | * | |
| Female | 57 | 4.0 (4.0) | 2.0 (2.0) | 1.0 (2.0) | 0.0 (1.0) | <0.001 | * | * | * | | * | |
| **Stroke subtype** | | | | | | | | | | | | |
| Ischemic stroke | 136 | 3.0 (3.0) | 1.0 (2.0) | 0.0 (1.0) | 0.0 (1.0) | <0.001 | * | * | * | * | * | |
| Hemorrhagic stroke | 16 | 4.0 (3.0) | 2.0 (3.0) | 1.5 (4.0) | 1.0 (4.0) | <0.001 | * | * | * | | | |
| **Stroke severity** | | | | | | | | | | | | |
| No stroke symptoms | 28 | 0.0 (1.0) | 0.0 (1.0) | 0.0 (0.0) | 0.0 (0.0) | <0.001 | | | * | | | |
| Minor stroke | 77 | 2.0 (3.0) | 1.0 (2.0) | 0.0 (1.0) | 0.0 (1.0) | <0.001 | * | * | * | * | * | |
| Moderate stroke | 37 | 4.0 (2.0) | 2.0 (2.0) | 2.0 (3.0) | 1.0 (3.0) | <0.001 | | * | * | * | * | |
| Severe stroke | 10 | 5.0 (0.0) | 4.5 (1.0) | 4.5 (1.0) | 4.0 (2.0) | 0.012 | | | | | | |

No stroke symptoms, NIHSS 0; Minor stroke, NIHSS 1–4; Moderate stroke, NIHSS 5–15; Severe stroke, NIHSS 16–42.

P-value of mean rank difference using Friedman Test; IQR = Interquartile range.

*Post hoc-test with p-value <0.05 for mRS differences between two-time points using Bonferroni correction (0 = at discharge, 1 = at one-month post-discharge, 3 = at three-month post-discharge, and 6 = at six-month post-discharge.

**Table 3. Barthel index scores at discharge, at 1-month, at 3-month and at 6-month post-discharge.**

| Variables | Discharge n | Mean (SD) | 1 month n | Mean (SD) | 3 months n | Mean (SD) | 6 months n | Mean (SD) |
|---|---|---|---|---|---|---|---|---|
| **All** | 155 | 70.6 (28.5) | 155 | 89.6 (23.8) | 153 | 92.9 (19.8) | 152 | 93.1 (20.4) |
| **Gender** | | | | | | | | |
| Male | 97 | 75.1 (27.3) | 97 | 92.2 (21.7) | 96 | 94.1 (18.2) | 95 | 94.7 (17.6) |
| Female | 58 | 63.0 (28.9) | 58 | 85.3 (26.6) | 57 | 90.8 (22.2) | 57 | 90.3 (24.3) |
| **Stroke subtype** | | | | | | | | |
| Ischemic stroke | 139 | 73.4 (26.8) | 139 | 90.8 (22.9) | 137 | 94.0 (18.4) | 136 | 93.9 (19.4) |
| Hemorrhagic stroke | 16 | 45.9 (31.1) | 16 | 78.7 (28.5) | 16 | 83.4 (28.4) | 16 | 85.6 (27.2) |
| **Stroke severity** | | | | | | | | |
| No stroke symptoms | 28 | 93.7 (16.4) | 28 | 99.8 (0.9) | 28 | 100.0 (0.0) | 28 | 100.0 (0.0) |
| Minor stroke | 79 | 78.9 (19.6) | 79 | 97.7 (6.9) | 78 | 99.0 (5.0) | 77 | 99.4 (3.6) |
| Moderate stroke | 37 | 50.3 (25.2) | 37 | 81.3 (27.9) | 37 | 86.9 (25.3) | 37 | 86.6 (27.0) |
| Severe stroke | 11 | 20.9 (20.9) | 11 | 32.7 (30.3) | 10 | 47.5 (30.4) | 10 | 48.5 (32.9) |

No stroke symptoms, NIHSS 0; Minor stroke, NIHSS 1–4; Moderate stroke, NIHSS 5–15; Severe stroke, NIHSS 16–42.

Table 4 displays seven models estimated from linear mixed models based on a combination of factors: the measurement of time (Model 1), time + gender (Model 2), time + age (Model 3), time + stroke subtype (Model 4), time + stroke severity (Model 5), time + gender + age + stroke subtype + stroke severity (Model 6), and time + gender + stroke subtype + stroke severity (Model 7). The adjusted Barthel index scores showed increasing trends from discharge to six months post-discharge in all models. However, significant changes were identified only at the time of discharge and one month

**Table 4. The linear mixed models with the main effect covariates.**

| Model | Covariates | Levels | Beta | SE | P-value |
|---|---|---|---|---|---|
| 1 | Time | Discharge | 66.9 | 4.9 | <0.001 |
| | | 1 month | 22.4 | 2.6 | <0.001 |
| | | 3 months | 3.5 | 2.6 | 0.192 |
| | | 6 months | 0.2 | 2.7 | 0.945 |
| 2 | Time + Gender | Discharge | 69.4 | 4.9 | <0.001 |
| | | 1 month | 22.4 | 2.6 | <0.001 |
| | | 3 months | 3.5 | 2.6 | 0.187 |
| | | 6 months | 0.2 | 2.6 | 0.940 |
| | | Female | −6.7 | 1.9 | 0.001 |
| 3 | Time + Age | Discharge | 80.3 | 6.9 | <0.001 |
| | | 1 month | 22.4 | 2.6 | <0.001 |
| | | 3 months | 3.4 | 2.6 | 0.193 |
| | | 6 months | 0.2 | 2.6 | 0.950 |
| | | Age | −0.21 | 0.1 | 0.006 |
| 4 | Time + Stroke subtype | Discharge | 68.4 | 4.9 | <0.001 |
| | | 1 month | 22.5 | 2.6 | <0.001 |
| | | 3 months | 3.3 | 2.6 | 0.180 |
| | | 6 months | 0.2 | 2.6 | 0.941 |
| | | HS | −14.6 | 3.0 | <0.001 |
| 5 | Time + Stroke severity | Discharge | 81.8 | 3.6 | <0.001 |
| | | 1 month | 22.2 | 1.9 | <0.001 |
| | | 3 months | 3.3 | 1.9 | 0.087 |
| | | 6 months | 0.2 | 1.9 | 0.922 |
| | | NIHSS | −3.5 | 0.1 | <0.001 |
| 6 | Time + Gender + Age + Stroke subtype + Stroke severity | Discharge | 86.3 | 4.94 | <0.001 |
| | | 1 month | 22.2 | 1.88 | <0.001 |
| | | 3 months | 3.2 | 1.88 | 0.085 |
| | | 6 months | 0.2 | 1.88 | 0.920 |
| | | Female | −3.9 | 1.39 | 0.005 |
| | | Age | −0.1 | 0.05 | 0.356 |
| | | HS | −4.1 | 2.35 | 0.078 |
| | | NIHSS | −3.5 | 0.15 | <0.001 |
| 7 | Time + Gender + Stroke subtype + Stroke severity | Discharge | 83.2 | 3.6 | <0.001 |
| | | 1 month | 22.2 | 1.9 | <0.001 |
| | | 3 months | 3.2 | 1.9 | 0.085 |
| | | 6 months | 0.2 | 1.9 | 0.919 |
| | | Female | −4.1 | 1.4 | 0.003 |
| | | HS | −4.5 | 2.3 | 0.055 |
| | | NIHSS | −3.6 | 0.1 | <0.001 |

Age = Age in years; Gender = Female vs. male; Stroke subtype = Hemorrhagic stroke (HS) vs. ischemic stroke (IS); Stroke severity = National Institute Health Stroke Scale (NIHSS) score at discharge.

p-value obtained from comparing the log-likelihood of Model 6 and Model 7 = 0.282.

p-value obtained from comparing the log-likelihood of Model 7 and Model 8 (Time + Gender + Stroke severity) = 0.032.

post-discharge but not at three months and six months after discharge. Female stroke participants had significantly lower adjusted mean BI scores than male participants in Model 2 (Beta = −6.7), Model 6 (Beta = −3.9), and Model 7 (Beta = −4.1). Age is inversely related to the BI score in Model 3 (Beta = −0.21). The adjusted mean BI scores for hemorrhagic stroke patients were lower than ischemic stroke patients in Model 4 (Beta = −14.6), Model 6 (Beta = −4.1), and Model 7 (Beta = −4.5). The severity of stroke showed adverse effects on the BI scores. One increasing NIHSS score reduced BI scores by 3.5 and 3.6 in Models 5 and 7, respectively. In Model 6, age was found to be insignificant (p = 0.356) when factors of gender, stroke subtype, and stroke severity were controlled. Then, we removed the age variable from the model (Model 7). The log-likelihood change after removing the age variable was not significant (p = 0.282). In Model 7, the effect of the stroke subtype on BI scores became borderline (p = 0.055). However, the log-likelihood change when removing the stroke subtype from the model was significant (p = 0.032). Therefore, the stroke subtype was retained in the final model (Model 7) for estimating BI scores among post-stroke participants.

## Discussion

This study used the modified Rankin scale (mRS) and Barthel index (BI) scales to examine patterns of change in functional outcomes in stroke patients over six months. Due to each measurement's limitations, both instruments were used. The Barthel index is an objective scale used to evaluate a patient's actual ability in activities of daily living (ADLs); nevertheless, it has limitations in demonstrating changes in functional outcomes in individuals with minor stroke or no stroke symptoms who scored 100 on the BI [18,25]. The modified Rankin scale was also used to assess patients' ability to perform their usual duties and activities before the stroke. However, the mRS is a subjective measure with no clear criteria for scoring [18,22]. Therefore, we employed both mRS and BI scales to better explain functional outcomes in stroke patients and improve measurement reliability [18,22,25].

As a result, functional outcomes significantly improved for stroke patients between discharge and one, three, and six months later. However, these functional results appeared to remain stable after three months post-discharge. These results were consistent with the findings from previous studies [9,10,15,20,28]. Neural plasticity is a significant mechanism for motor recovery following stroke [29,30]. Plasticity refers to the brain's ability to reorganize itself in response to motor impairments [29,30]. Motor recovery following a stroke is time-dependent and influenced by several factors. The most significant improvements occur in the first few weeks following a stroke, with the greatest possible recovery occurring within three months, with less recovery after three months [29–34]. Early rehabilitation or motor training within two weeks has been shown to improve motor recovery and functional outcomes [35].

Functional outcomes in stroke patients differed by gender, stroke subtype, and stroke severity. From the time of discharge to 6 months after discharge, female stroke patients appeared to have worse functional outcomes than male patients. The result from linear mixed models revealed that females had lower BI scores than males by 4.1 scores while adjusted for time after stroke, stroke subtypes, and stroke severity. This finding was inconsistent with previous studies from Malaysia [15] and Sweden [6]. A previous study in Malaysia found that female stroke patients had lower BI scores than males at discharge, one month, and three months after discharge [15]. However, they found no significant effect of gender on BI improvement in their adjusted models [15]. Furthermore, another study in Sweden found that females had a greater proportion of ADL dependence at three months in univariate analysis but did not find a significant impact of gender in multivariate models [6]. It might be concluded that Thai females who had a stroke tended to have poorer functional outcomes than males. Further research is needed to confirm the conclusions of this study. Female stroke patients may require extra support due to having poorer functional outcomes.

This study found that individuals with hemorrhagic stroke had poor functioning outcomes within six months after discharge. Hemorrhagic stroke patients had poorer functional outcomes (mRS ≥ 3) compared to ischemic stroke patients at discharge (62.6% vs. 52.2%), one-month post-stroke (37.5% vs. 17.9%), three-month post-stroke (31.2% vs. 12.2%), and six-month post-stroke (25.0% vs. 10.9%). Unfortunately, previous studies measuring functional outcomes using mRS did

not provide mRS distribution by stroke subtype [9,10]. In this recent study, hemorrhagic stroke patients showed lower BI than ischemic stroke patients by 14.6 scores. This finding was consistent with a previous study in Malaysia, which found that hemorrhagic stroke patients had a 16.8-point lower BI than those who had ischemic stroke when age and time were controlled [15]. Moreover, when we included the variables of age and stroke severity in the model, we found that hemorrhagic stroke patients had significantly lower BI than ischemic stroke patients by 4.1 scores. Hemorrhagic stroke patients had poorer functional outcomes than ischemic stroke patients [15].

In addition, this study indicated that functional outcome improvement significantly differed by stroke severity. Patients who had a minor stroke were more likely to have a good recovery in functional outcomes than those who had a more severe stroke. Patients with a mild or moderate stroke had significant improvements in mRS scores. Patients with a severe stroke after treatment had minor changes in their functional outcomes from discharge to 6 months. However, 72.8% of severe stroke patients continued to suffer moderately severe to severe disability (mRS ≥ 4). Moreover, this study identified that every NIHSS score decreased BI by 3.6 points. The results were consistent with previous studies suggesting that more stroke severity led to poorer functional outcomes [6,9,10,30,36].

In the study setting, multidisciplinary rehabilitation teams, including physical therapy, occupational therapy, speech therapy, or other related healthcare professionals, were consulted for each patient based on each patient's specific problems and needs. Specific rehabilitation programs were developed for each stroke patient who had neurological impairments. Before discharge, patients with neurological impairment received a personalized rehabilitation plan. They were instructed on how to follow the plan themselves or with the help of family members at home and they were also scheduled for follow-up appointments with rehabilitation specialists.

In conclusion, improvements in functional outcomes in stroke patients significantly differed by time, gender, stroke subtype, and stroke severity. Female and hemorrhagic stroke patients had poorer functional outcomes compared to male and ischemic stroke patients. Patients with a minor stroke had the most potential to recover, followed by those with a moderate stroke. Early implementation of rehabilitation and supportive programs is recommended to help stroke patients achieve optimal functional recovery. Specific assessments on stroke burden and supportive programs are suggested for females, hemorrhagic stroke, and more severe stroke patients. Patients who experienced a severe stroke had poor functional outcomes and were more likely to live with chronic disability and rely on others. Family members or caregivers should be prepared to care for patients discharged with a severe stroke. Severe stroke patients also require rehabilitation and appropriate care to improve their quality of life and prevent complications [23,37–39].

The strengths of this study include a follow-up period of up to six months after a stroke. This timeframe covers the crucial motor recovery phase, which typically occurs within the first three months [29–34]. Additionally, it allows for further improvement within six months, as spontaneous recovery generally reaches its peak by that time [20,30]. Moreover, this is the first analysis of longitudinal stroke data in Thailand to identify patterns of change in functional outcomes and their prognostic factors.

Some limitations should be acknowledged in this study. First, the small sample size from a single study setting may limit the generalization of the study findings. A larger sample size is needed to enhance the generalizability of the results and improve the statistical power of the analyses. Second, this study only included cognitively intact patients. Those with more severe strokes, cognitive impairments, or communication difficulties due to aphasia were excluded, which may have led to an overestimation of functional outcomes. Third, more data on other factors, such as depression, are necessary to provide more valid results on functional outcomes following stroke [40]. Forth, telephone interviews had a tendency to yield incomplete information. However, telephone interviews offer a simple way of assessing stroke patients' functional outcomes with less lost follow-up. Lastly, this study did not provide information on stroke rehabilitation programs for stroke patients. Future research should collect and analyze data related to multidisciplinary rehabilitation approaches, as it may influence the functional improvement of acute stroke patients.

## Supporting information

**S1 Table. Frequencies and percentages of modified Rankin scale (mRS) scores.**
(DOCX)

## Acknowledgments

We would like to express our gratitude to the stroke patients who participated in the study.

## Author contributions

**Conceptualization:** Nipaporn Butsing, Nalinrat Thongniran, Jesada Keandoungchun.

**Data curation:** Nipaporn Butsing, Nalinrat Thongniran.

**Formal analysis:** Nipaporn Butsing, Nalinrat Thongniran.

**Funding acquisition:** Nipaporn Butsing.

**Investigation:** Nipaporn Butsing, Nalinrat Thongniran.

**Methodology:** Nipaporn Butsing, Jesada Keandoungchun.

**Project administration:** Nipaporn Butsing.

**Resources:** Nipaporn Butsing, Nalinrat Thongniran, Jesada Keandoungchun.

**Software:** Nipaporn Butsing.

**Supervision:** Nipaporn Butsing, Jesada Keandoungchun.

**Validation:** Nipaporn Butsing, Nalinrat Thongniran, Jesada Keandoungchun.

**Visualization:** Nipaporn Butsing.

**Writing – original draft:** Nipaporn Butsing.

**Writing – review & editing:** Nipaporn Butsing, Nalinrat Thongniran, Jesada Keandoungchun.

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
