## [Decision Letter · Decision Letter 0]

27 Jan 2025

Dear Dr. Butsing,

Thank you for submitting your manuscript to PLOS ONE. After careful consideration, we feel that it has merit but does not fully meet PLOS ONE’s publication criteria as it currently stands. Therefore, we invite you to submit a revised version of the manuscript that addresses the points raised during the review process.

We look forward to receiving your revised manuscript.

Kind regards,

Elvan Wiyarta, M.D.

Academic Editor

PLOS ONE

Journal Requirements:

2. In the online submission form, you indicated that [Data cannot be shared publicly because of the institution's data-sharing policy, as it contains patients' clinical information. Data is available upon request and ethics committee approval. Researchers may submit their research proposals to the Human Research Ethics Committee, Faculty of Medicine Ramathibodi Hospital at https://www.rama.mahidol.ac.th or contact the corresponding author for help.].

Reviewers' comments:

Reviewer's Responses to Questions

**Comments to the Author**

1. Is the manuscript technically sound, and do the data support the conclusions?

Reviewer #1: Yes

Reviewer #2: Partly

2. Has the statistical analysis been performed appropriately and rigorously?

Reviewer #1: Yes

Reviewer #2: Yes

3. Have the authors made all data underlying the findings in their manuscript fully available?

Reviewer #1: Yes

Reviewer #2: Yes

4. Is the manuscript presented in an intelligible fashion and written in standard English?

Reviewer #1: Yes

Reviewer #2: No

Reviewer #1: This cohort describes factors of changes in functional outcome after a first-time stroke. The results of this paper show lots of outstanding outcomes of functional recovery after stroke and discharge. Functional status recovery after an acute stroke varies significantly depending on factors such as the severity and type of stroke, the area of the brain affected, the individual’s pre-stroke health, and the medical intervention such as medications and rehabilitation. Although I appreciate such an achievement, I still have three major questions and those questions that authors should reveal key points that are included in the method.

1. Location and size of the stroke in the brain imaging by CT, MRI, or angiography.

2. Timing of treatment such as thrombolysis with tPA or mechanical thrombectomy or supportive care only.

3. Any multidisciplinary rehabilitation including physical therapy, occupational therapy, and speech therapy.

Reviewer #2: Authors report an interesting analysis of stroke outcome in Thailand, pointing out the profound differences between minor and severe strokes, the worse outcome of ICH and longer F.U. time needed for ICH, etc. However, some flaws arise, as follows.

The selection process is not clearly described. For example, at line 103 - “The primary study included 186 consecutive acute first-stroke patients hospitalized…” - the primary study is not cited and is not possible to piece together the origin of this study cohort.

It also seems that Authors had “a priori” excluded fatal strokes and those patients who died later during the F.U. period (or mRS = 6), but this point not clear. I would be quite impressed that no death occurred at least within hemorrhagic and severe strokes. More, they state anyway that one patient died before 6 months F.U., and this contributes to make unclear how they managed fatal strokes. Overall, this point should be addressed/explained in M&M and in the “sample selection process flow chart (figure 1). It could be mentioned in the Discussion as well.

Furthermore, the exclusion of aphasic patients should be at least reported as a limitation since aphasia is one of the most important neurological deficits after stroke.

Authors did not mention/consider rehabilitation regimens (i.e., intensive, extensive, outpatients’ programs, no rehab at all, etc.) in their analysis. At least the proportion of patients who underwent an active intervention to modify functional outcome could be reported.

The following two sentences (lines 232-235) are hard reading, and they seem contradictory, please clarify: “Participants with hemorrhagic stroke had poorer functional outcomes than those with ischemic stroke, with significant changes in mRS median scores from discharge to 1 month, 3 months, and 6 months post-discharge (p < 0.05). However, there was no significant change in mRS scores from 1-month to 3-month after discharge from hemorrhagic stroke”.

English phrasing must be revised in several sections. Following, only very few examples:

Line 25 - and varied by according to/depending on several factors…

Line 56 - worldwide the world [1].

Line 80 – 3 months…

Within Figure 1: all strokes

**Do you want your identity to be public for this peer review?** For information about this choice, including consent withdrawal, please see our Privacy Policy

Reviewer #1: No

Reviewer #2: **Yes: ** Francesco Janes

---

## [Author Response · Author response to Decision Letter 1]

16 Feb 2025

Dear Editor and reviewers,

Thank you for your comments and the reviewers' suggestions on our manuscript, "Changes in functional outcome after a first-time stroke: Data from a longitudinal study" (ID PONE-D-24-52408). All authors are very grateful for the constructive and valuable feedback from the two reviewers, which has significantly helped us improve this revised version of the manuscript.

Please find our point-by-point responses to the reviewers' comments uploaded in the submission system. In accordance with the instructions in your letter, we have uploaded a rebuttal letter that addresses each point raised by the academic editor and reviewers in the 'Response to Reviewers' section. Additionally, we have submitted a clean version of the revised manuscript under the 'Manuscript' section, along with a highlighted version as a separate file in the 'Revised Manuscript with Track Changes' section.

Furthermore, we have added more information to Figure 1, corrected a typo in Figure 2, and uploaded the new versions of both figures to the system. Moreover, the related manuscript was also uploaded. If you have any questions, please feel free to let me know.

Sincerely,

Nipaporn Butsing, RN, DrPH

Assistant Professor, Ramathibodi School of Nursing

Faculty of Medicine Ramathibodi Hospital, Mahidol University, Bangkok, Thailand.

Email: nipaporn.but@mahidol.edu

---

## [Decision Letter · Decision Letter 1]

27 Jul 2025

Changes in functional outcome after a first-time stroke: Data from a longitudinal study

PONE-D-24-52408R1

Dear Dr. Butsing,

We’re pleased to inform you that your manuscript has been judged scientifically suitable for publication and will be formally accepted for publication once it meets all outstanding technical requirements.

Kind regards,

Elvan Wiyarta, M.D.

Academic Editor

PLOS ONE

Additional Editor Comments (optional):

Reviewers' comments:

Reviewer's Responses to Questions

**Comments to the Author**

Reviewer #2: All comments have been addressed

2. Is the manuscript technically sound, and do the data support the conclusions?

Reviewer #2: Yes

3. Has the statistical analysis been performed appropriately and rigorously?

Reviewer #2: Yes

4. Have the authors made all data underlying the findings in their manuscript fully available?

Reviewer #2: Yes

5. Is the manuscript presented in an intelligible fashion and written in standard English?

Reviewer #2: Yes

Reviewer #2: (No Response)

**Do you want your identity to be public for this peer review?** For information about this choice, including consent withdrawal, please see our Privacy Policy

Reviewer #2: **Yes: ** Francesco Janes
